# Long Noncoding RNAs in Acute Myeloid Leukemia: Functional Characterization and Clinical Relevance

**DOI:** 10.3390/cancers11111638

**Published:** 2019-10-24

**Authors:** Morgane Gourvest, Pierre Brousset, Marina Bousquet

**Affiliations:** Cancer Research Center of Toulouse (CRCT), UMR1037 INSERM—Université Paul Sabatier Toulouse III—CNRS ERL5294, 31037 Toulouse, France; morgane.gourvest@inserm.fr (M.G.); brousset.p@chu-toulouse.fr (P.B.)

**Keywords:** long noncoding RNA, acute myeloid leukemia, biomarkers

## Abstract

Acute Myeloid Leukemia (AML) is the most common form of leukemia in adults with an incidence of 4.3 per 100,000 cases per year. Historically, the identification of genetic alterations in AML focused on protein-coding genes to provide biomarkers and to understand the molecular complexity of AML. Despite these findings and because of the heterogeneity of this disease, questions as to the molecular mechanisms underlying AML development and progression remained unsolved. Recently, transcriptome-wide profiling approaches have uncovered a large family of long noncoding RNAs (lncRNAs). Larger than 200 nucleotides and with no apparent protein coding potential, lncRNAs could unveil a new set of players in AML development. Originally considered as dark matter, lncRNAs have critical roles to play in the different steps of gene expression and thus affect cellular homeostasis including proliferation, survival, differentiation, migration or genomic stability. Consequently, lncRNAs are found to be differentially expressed in tumors, notably in AML, and linked to the transformation of healthy cells into leukemic cells. In this review, we aim to summarize the knowledge concerning lncRNAs functions and implications in AML, with a particular emphasis on their prognostic and therapeutic potential.

## 1. Introduction

Acute myeloid leukemia (AML) is an aggressive malignancy of the hematopoietic system caused by malignant transformation of hematopoietic stem- or progenitor cells. AML is the most common form of acute leukemia in adults with an incidence of 4/100,000 cases per year [1]. Hallmarks of Acute Myeloid Leukemia are an uncontrolled proliferation and subsistence of myeloid progenitors that are blocked in differentiation stages. Even though one-half of patients have chromosomal abnormalities, the other half are cytogenetically normal (CN-AML), carrying recurrent somatic mutations in several oncogenes (NPM1, FLT3, CEBPA) [2]. Historically, the identification of genetic alterations in AML focused on protein-coding genes to provide prognostic tools and to understand the molecular complexity of AML [3]. Despite these findings and because of the heterogeneity of this disease, questions as to the molecular mechanisms underlying AML development and progression remained unsolved. High throughput technologies (tilling arrays, next-generation sequencing) demonstrated pervasive transcription and estimate that up to 90% of the Human Genome may be transcribed as noncoding RNAs (ncRNAs) [4,5,6]. Even though small ncRNAs were extensively characterized (rRNAs, tRNAs, snoRNAs, miRNAs), profiling also highlighted long noncoding RNAs (lncRNAs). Increasing evidence indicates that lncRNAs play an important role in various cellular processes by regulating the different steps of gene expression [7,8,9,10,11]. LncRNAs deregulation has been associated with malignant transformation and identified as potential biomarkers in various cancers [12]. Their implications in AML have only been highlighted recently but are steadily on the increase. Herein, we review the current understanding of lncRNAs deregulation in AML with an emphasis on their putative therapeutic potential. 

## 2. Background on lncRNAs

Arbitrary defined as transcripts of over 200 nucleotides without coding potential, lncRNAs were originally considered as dark matter due to their low expression and poor conservation across species compared to their messenger RNAs counterparts [13,14,15,16]. However, lncRNAs are tightly controlled and exhibit higher tissue and development specific expression than proteins—including lineage determining transcription factors—which support their biological functions [17,18]. Mostly transcribed by RNA Polymerase II, lncRNAs mimic mRNAs in their biogenesis and regulation: most of them are capped, polyadenylated, and spliced via canonical genomic splice site motifs [16]. Their transcription is also regulated by chromatin modifying complexes and specific transcription factors. Nevertheless, we also observe diversity with some non-polyadenylated or unspliced lncRNAs, lncRNAs transcribed by RNA Polymerase III [19,20] and snoRNA-related lncRNAs (sno-lncRNAs) expressed from introns via the snoRNP machinery [21].

Because of their heterogeneity, much debate remains to find the best classification system for lncRNAs that are currently defined according to their genomic location. Indeed, lncRNAs can be divided into sense [22] or antisense [23] lncRNAs when the lncRNA sequence overlaps with the sense or antisense strand of a protein coding gene, respectively. They can also be defined as intronic lncRNA when it is derived entirely from an intron of another gene, as bidirectional when the transcription of the lncRNA is initiated in close proximity (<1kb) and opposite orientation to a protein coding gene [24], or as intergenic when the lncRNA is not located near any other protein coding genes [25]. In addition, lncRNAs can also be produced from enhancer (eRNAs) or promoter regions (pRNA or PROMPTs) [26]. 

As illustrated in Figure 1, lncRNAs are also a multifaced family in terms of compartmentalization and mechanisms of action. It is well now established that lncRNAs functions are directly associated with their subcellular fates [27]. They can be nuclear, cytoplasmic, or both, located in subcellular compartments such as nuclear bodies [28] or excreted into exosomes [29]. They positively or negatively regulate the different steps of gene expression by their ability to bind DNA, RNA, and proteins. 

Nuclear lncRNAs are implicated in epigenetic and transcriptional regulations by recruiting activator or repressor chromatin-modifying complexes and transcription factors onto target genes [30,31]. Most of the studies have been focused on lncRNAs and Polycomb repressive complex 2 (PRC2) showing that several lncRNAs such as Xist were able to recruit and guide PRC2 to regions of interest. However, the lncRNAs field is moving away from this perspective and some paper delivered contradictory results [32]. LncRNAs can act at their transcription site affecting the expression of nearby genes (cis) or can be relocated on different chromosomes to regulate the expression of distal genes (trans). LncRNAs can form a DNA-RNA duplex/triplex that anchors associated effectors to active chromatin sites such as promoters or enhancers [33]. Of note, the lncRNA chromatin binding pattern is not restricted to these regions and lncRNAs can also be found linked to 3’ part of genes [34]. LncRNAs are also involved in chromatin remodeling by forming inter/intra chromosomal loops [31,35]. LncRNAs are implicated in post-transcriptional modifications regulating either splicing or editing in the nucleus, and stability or translation of targeted messengers for cytoplasmic lncRNAs (Figure 1) [36]. Indeed, cytoplasmic lncRNAs can be microRNA sponges also called competitive endogenous RNAs (ceRNAs), containing microRNA binding sites [37]. MicroRNA sponges are able to sequestrate microRNAs and to keep them away from their messengers RNA targets leading to the stabilization of their targets. Finally, proteomic analyses revealed that a subset of transcripts annotated as lncRNAs may encode small micropeptides that may be functional [38,39,40,41].

Regulation of gene expression is crucial during hematopoietic differentiation [42]. LncRNAs seem to be key players controlling these different steps, from the maintenance of hematopoietic stem cells (HSC), the determination of their fate to the differentiation of progenitor and precursor blood cells [43,44]. Their expression seems to be specific to distinct hematopoietic cell types [45]. Due to their impact on these crucial steps, we understand that lncRNAs deregulation could contribute to the development and progression of hematological malignancies such as acute myeloid leukemia (AML). This is supported by the fact that lncRNAs have already been found as upregulated or downregulated in several types of tumors and could exert oncogenic or tumor suppressor functions [46]. Moreover, lncRNAs expression has already been correlated with diagnostic and prognostic factors and their incorporation in clinical routine is starting to be considered. Cancer biomarkers currently available in the clinic for cancer treatment are mostly based on protein coding genes, immunohistochemistry methods allowing easy detection of deregulated protein levels or oncogenic proteins between normal and cancer cells. However, high specificity and non-invasive detection of lncRNAs in tissues, serum, plasma, urine, and saliva support the fact that they could be promising predictive biomarkers and potential therapeutic targets in cancer. 

Concerning the potential role of lncRNAs as diagnostic biomarkers, the most inspiring example is PCA3 (Prostate Cancer Antigen 3) for prostate cancer diagnosis. It is a prostate specific lncRNA overexpressed in prostate cancer patients. Found in the urine of most cancer patients compared to healthy patients, it is undetectable in other tumor types [47,48]. Although its biological function is unclear, a urinary PCA3 detection kit was developed and approved by the US Food and Drug Administration (FDA) for prostate cancer diagnosis [49,50]. Several other lncRNAs have been proposed for the diagnosis of various cancer types [51]. The upregulation of HULC which occurs in hepatocellular carcinomas has also been observed in patient blood serum [52]. LncRNA HOTAIR was strongly proposed as a peripheral blood diagnostic biomarker in thyroid cancer [53]. However, no other lncRNA-based diagnostic tool has been developed for widespread use, nor for AML diagnosis.

## 3. Regulatory Roles of lncRNAs in AML

In AML, several lncRNAs have been characterized and described as oncogenes (Table 1) or tumor suppressors (Table 2). To exert their functions, lncRNAs have multiple mechanisms of action. Over the past decades, the growing characterization of lncRNAs in leukemogenesis highlighted the diversity of lncRNA mechanisms. Non-exhaustive, this review summarizes the functions of the best characterized lncRNAs that have been shown to be deregulated in AML and play regulatory roles in AML pathogenesis. (Figure 2)

The best described function of lncRNAs is their involvement in epigenetic and transcriptional regulation. They are known to act as guides, leading to the recruitment of proteins or complexes to chromatin sites to regulate downstream gene expression [30,31]. 

For example, Pasmant et al. identified ANRIL (Antisense ncRNAs in the INK4 locus) as an antisense lncRNA from the *p15^INK4b^* locus [54]. Kotake et al. demonstrated that ANRIL silenced in cis *p15^INK4b^* by recruiting the repressive complex PRC2 on its promoter [55,56]. ANRIL has been shown to be up-regulated in various cancers, namely in AML primary samples [57]. In AML, ANRIL promotes malignant cell survival and AML progression by regulating glucose metabolism. ANRIL represses the expression of Adiponectin receptor (AdipoR1), a key regulator of the glucose metabolism which regulates AMPK and SIRT1 phosphorylation level [57]. Unfortunately, the underlying mechanisms have not been studied.

Other examples of lncRNAs involved in epigenetic regulation are lncRNAs located in the HOX gene cluster. Indeed, numerous lncRNAs transcribed within the HOX cluster have been discovered to tightly cooperate and regulate the expression of this region [58]. The deregulation of the HOX gene cluster has long been regarded as an important mechanism of leukemogenesis [59].

HOTAIR (HOX Transcript Antisense RNA), a lncRNA expressed from the HOXC locus on chromosome 12, is historically recognized as a trans-acting repressor of genes in the HOXD locus, by guiding PRC2 complexes on their target promoters. Its overexpression has been documented in various human solid tumors [60,61]. HOTAIR is also upregulated in various AML cell lines [62,63]. In vitro knock-down of HOTAIR inhibited cell growth, induced apoptosis, and reduced the number of colony formation units. HOTAIR represses in trans p15 expression through H3K27 tri-methylation of its promoter mediated by PRC2 in AML [63]. HOTAIR also recruits DNMT3B to increase HOXA5 promoter methylation. Knock-down of HOTAIR and consequently upregulation of HOXA5 were found to induce apoptosis and reduce the proliferation of AML cells in vivo [64]. 

Transcribed from the intergenic and antisense region of the HOXA cluster gene, HOTAIRM1 (HOXA Transcript Antisense RNA Myeloid-Specific 1) is also one of the most studied myeloid lncRNA and its expression seems to be restricted to myeloid cells [65]. It is upregulated during ATRA-induced-granulocytic differentiation of NB4 promyelocytic leukemia cells and primary human hematopoietic cells [65]. HOTAIRM1 enhances in cis the expression of its neighboring HOXA1-4 genes and crucial myeloid differentiation CD11b,c and CD18 markers after ATRA treatment through the recruitment of the UTX/MLL epigenetic complex on their promoters [65,66]. Knocking down HOTAIRM1 in the NB4 cell line retarded induced granulocytic differentiation, resulting in a significantly larger population of immature and proliferating cells that maintained cell-cycle progression from G1 to S phases [67]. These data suggest an oncogenic role of HOTAIRM1 in AML by regulating a switch from a proliferative phase toward granulocytic maturation. 

LncRNAs are also able to manage chromatin conformation and 3D shape of chromosomes to regulate gene expression. 

For example, RUNX1, a major transcription factor in hematopoiesis, is frequently disrupted by mutations or chromosomal translocations in AML. One of the most common is the t(8;21) translocation found in 30–40% of AML cases producing a RUNX1-ETO chimeric protein [68,69]. However, the specific molecular mechanism underlying the high frequency of chromosomal translocations of RUNX1 in human malignancies is unknown. An RNA-guided Chromatin Conformation capture has cleverly identified RUNXOR (RUNX1 overlapping RNA), a lncRNA overlapping the RUNX1 promoter [70]. Wang et al. proposed that RUNXOR could use its 3’-terminal fragment to orchestrate the formation of an intra-chromosomal loop between the RUNX1 promoter and enhancer (in cis), but also to participate in long-range interchromosomal interactions (in trans) with chromatin regions that are involved in multiple RUNX1 translocations and may influence translocation formation. The authors also showed that RUNXOR lncRNA could bind and regulate PRC2 complex and RUNX1 activity [70]. This could unveil a candidate involved in the formation of chromosomal translocations in hematopoietic malignancies. 

Sun et al. also cleverly identified IRAIN (IGF1R antisense imprinted non-protein coding RNA), an antisense lncRNA originating from an intron of insulin-like growth factor type I receptor (IGF1R) [71]. IGF1R is a receptor tyrosine kinase abundantly activated in leukemic cells, giving them proliferative and treatment resistance capacities, through IGF1R receptor-mediated activation of the PI3K/Akt signaling pathway. However, molecular mechanisms underlying IGF1R gene deregulation in cancer remain unclear. IRAIN is involved in the formation of long-range intrachromosomal interaction between the IGF1R promoter and a distant intragenic enhancer. Overexpression of the IRAIN lncRNA inhibits tumor cell migration, suggesting its tumor suppressor function. However, underlying mechanisms need to be further studied [72]. 

LncRNAs are also implicated in post-transcriptional and translational regulations. UCA1 (Urothelial Cancer Associated 1) was firstly described by Hughes et al. who studied the effects of a dominant negative isoform of CEBPA, known as CEBPA-p30 [73]. Genome-wide transcriptome analysis of K562 cells with inducible CEBPA-p30 identified the lncRNA UCA1 as the most upregulated transcript. In vitro, UCA1 promotes cell viability, migration, invasion, and reduces apoptotic processes suggesting its oncogenic functions in CN-AML [73]. In this study, the authors show evidence that UCA1 lncRNA acts at the translational level to regulate gene expression. It seems to titrate the hnRNP1 protein, which normally facilitates translation of p27^kip1^. However, UCA1 can also act as microRNA sponge by titrating the miR-125a [74]. UCA1 consequently leads to the upregulation of its messenger RNA targets: the glycolysis regulator HK2 and HIF1α, and participate to chemoresistance. It has also been proposed that UCA1 enhances cell proliferation and survival by sponging mir-126 which regulates RAC1 GTPase [75]. 

CCAT1 (Colon cancer-associated transcript-1) is another example of a microRNA sponge in AML [76]. By reducing miR-155 availability and consequently upregulating c-MYC expression in AML cells, CCAT1 inhibits monocytic differentiation and promotes cell growth in vitro [76]. During the past year, the number of publications depicting lncRNAs as microRNA sponges has significantly increased [77,78,79,80,81,82,83,84,85,86,87,88]. 

Finally, because of their ability to bind DNA, RNA and proteins, the same lncRNA can exert multiple functions and mechanisms of action to coordinate multilevel gene regulation. 

For example, despite its proven role as an epigenetic regulator, H19 was recently described as miR-19a-3p and miR-29a-3p sponges in hematopoiesis and AML context [81,89]. By trapping these microRNAs, H19 sustains leukemic cell proliferation and limits apoptosis by respectively regulating the expression of IDH2 [89] and Wnt/βcat effectors [81]. H19 is also able to impede telomerase activity in ATRA-treated APL cells by disrupting the telomerase complex assembly [90]. 

As multilayer regulators, HOTAIR and HOTAIRM1 are also microRNA sponges. HOTAIR is able to titrate miR-193a which targets c-KIT [62], while HOTAIRM1 sponges miR-20a, miR-106a and miR-125b which targets ULK1, E2F1, and DRAM2 respectively. This role of HOTAIRM1 leads to the degradation by autophagy of the PML-RARα chimeric oncoprotein found in APL [91]. In fine, the PML-RARα proteolysis induced by HOTAIRM1 restores promyelocytic differentiation.

Lastly, NEAT1 (Nuclear paraspeckle assembly transcript 1) that plays a major role in the formation of sub-nuclear paraspeckles, is overexpressed and associated with poor survival in many solid tumors where it influences the expression of downstream effectors by interacting with PRC2, acts as miRNA sponges and suppresses the expression of miR-129 by promoting its promoter DNA methylation [92]. Paradoxically, NEAT1 is dramatically downregulated in PML-RARα APL samples compared to healthy donors suggesting its role as a tumor suppressor [93]. In AML, NEAT1 downregulation has been demonstrated in vitro to be caused by PML-RARα and can be restored by ATRA treatment. In NB4 cells, in vitro silencing of NEAT1 blocks ATRA-induced differentiation [93]. Consistently, NEAT1 is highly expressed in stem and progenitor cells and significantly reduced in granulocytes [94]. Recently, Zhao et al. also demonstrated in vitro that NEAT1 competitively binds miR-23a-3p to regulate SMC1A expression in AML that consequently inhibits AML cell proliferation and induces cell-cycle arrest and apoptosis [95]. 

## 4. LncRNAs as Biomarker Candidates among Acute Myeloid Leukemia

### 4.1. Specific lncRNA Signatures among AML Subtypes

The large-scale analysis highlighted distinct lncRNAs expression patterns associated with specific AML subtypes, reflecting the heterogeneity of this disease. A pioneering study came from Garzon et al. in 2014, who investigated lncRNAs expression in a cohort of 148 older Cytogenetically Normal AML (CN-AML) patients using a custom microarray. They found distinctive lncRNAs profiles associated with recurrent mutations such as FLT3-ITD, NPM1, CEBPA, IDH2, ASLX1 or RUNX1 [120]. They identified signatures of lncRNAs associated with the mutational status of NPM1 and the presence of FLT3-ITD. In agreement with these results, De Clara et al. deciphered the lncRNAs transcriptome associated with recurrent AML mutations by performing RNA sequencing on CN-AML patients [113]. These results improved the overall knowledge of lncRNAs, as their RNAseq approach uncovered more than 8000 new lncRNAs. A major finding of this study was the discovery and the validation of a minimal set of 12 lncRNAs able to discriminate NPM1-mutated from NPM1 wild type patients. In addition, Diaz-Beya et al. studied lncRNAs expression by microarray in various AML cases associated with chromosomal abnormalities, with an emphasis on the t (8;16) translocation. They found a signature of lncRNAs differentially expressed in t(8;16) positive cases, in particular the upregulated linc-HOXA11, HOXA11-AS, HOTTIP, and NR_038120 [121]. Zhang et al. also uncovered a subset of lncRNAs enriched in Acute Promyelocytic Leukemia patients harboring the PML-RARα translocation [122]. Recently, Schwarzer et al. reported specific fingerprints of lncRNAs in various subgroups of pediatric AML and identified stem cell signatures in normal HSCs that were upregulated in AML blasts, highlighting stemness patterns of AML blasts [123]. 

Most of these profiling studies also intended to correlate these lncRNA signatures with treatment response and survival hoping to highlight putative prognostic biomarkers. Garzon et al. established a signature of 48 lncRNAs associated with event free survival (EFS) and were able to build a prognostic score based on the expression of these 48 survival-associated lncRNAs [120]. Patients with an unfavorable lncRNA score were found to have reduced complete remission (CR) rates following intensive chemotherapy, a reduced disease-free survival (DFS) and overall survival (OS) compared to patients with a favorable lncRNA score. Remarkably, the prognostic value of lncRNAs remained independent from other prognostic biomarkers in a multivariate analysis, suggesting that lncRNAs could refine risk stratification of CN-AML patients [113]. 

Concerning the rare population of leukemic stem cells (LSCs) assumed to be responsible for relapse because of their abnormal self-renewal capacity and increased chemotherapy resistance, Bill et al. identified an LSC-specific signature of 111 lncRNAs that correlated with a previously identified coding gene expression signature associated with LSCs [114]. 

Tsai et al. go further and propose the incorporation of lncRNA profiles in the 2017 EuropeanLeukemiaNet (ELN) risk classification [124,125]. They formulated a lncRNA scoring system based on the expression of five significant prognosticator lncRNAs. They validated this scoring system on 275 newly diagnosed AML patients. Patients with high scores had less favorable-risk cytogenetics and presented more gene mutations at diagnosis than low score patients. High score patients had lower CR, a trend of higher relapse rates, shorter OS and DFS. In the intermediate-risk subclass, they observed that patients with lower lncRNA scores had OS and DFS similar to favorable-risk cytogenetics patients, while patients with higher lncRNA scores had OS and DFS similar to those with unfavorable-risk cytogenetics. They also showed that hematopoietic stem cell transplantation did confer a benefit to the ELN intermediate-risk patients with higher lncRNA scores, but not to those with lower scores. The lncRNA scoring system could help to stratify the 2017 ELN intermediate-risk patients and provide guidance for the choice of treatment strategies.

The benefits of these extensive profiling studies were to identify new lncRNAs and highlight their prognostic potential. The functional and molecular characterization of these candidates could unravel their interactions with known driver events and to provide a better understanding as to how each factor can influence the development and progression of AML in order to unveil new potential therapeutic targets.

### 4.2. Prognostic lncRNA Biomarker Candidates

Most of in vitro characterized lncRNAs in AML has been investigated in patient samples to determine if they were associated with good or poor prognosis and drug resistance (Table 1 and Table 2).

#### 4.2.1. Good Prognostic lncRNA Biomarkers

As in leukemia cell lines, IRAIN expression is downregulated in AML patients. Lower IRAIN is associated with high-risk AML patients and higher IRAIN with the low-risk group [71,72]. Low expression of IRAIN is independently associated with an adverse prognosis: higher WBC and blast counts and shorter OS and DFS [72]. Besides, patients with refractory response to chemotherapies and those with relapse were more likely to show a lower initial IRAIN expression. 

The lncRNA CASC15 (cancer susceptibility candidate 15) is upregulated in pediatric RUNX1-rearranged AML, with the highest level found in AML with the t(8;21) translocation and is associated with a good prognosis [119]. Overexpression of CASC15 increased apoptosis, myeloid differentiation and decreased engraftment of primary bone marrow cells in the hematopoietic system, suggesting its antileukemic role. Paradoxically, CASC15 positively regulates expression of its adjacent gene, SOX4, thought to function as an oncogene in AML. 

The maternally expressed 3 lncRNA (MEG3) is a tumor suppressor lncRNA downregulated in various solid tumors [126,127,128]. In AML, Benetatos et al. found that MEG3 hypermethylation occurred in 48% of AML cases and conferred a significantly reduced OS rate in these patients [115]. Hypermethylation of the MEG3 promoter was confirmed by several studies [116,117]. However, Sellers et al. examined mononuclear cells of AML patients and determined that patients with increased methylation at this locus had a significantly longer OS than patients with lower methylation at this locus. The authors explained this disparity by differences in methodology and CpG site selection across studies. Merkerova et al. also confirmed that MEG3 hypermethylation was associated with longer DFS [118]. Further studies of MEG3 in AML is imperative for a greater understanding of its role in pathogenesis.

#### 4.2.2. Poor Prognostic lncRNA Biomarkers

Located upstream of the CDKN1A promoter, PANDAR (Promoter of CDKN1A Antisense DNA Damage Activated RNA) is involved in cell proliferation, migration, invasion, and apoptosis of cancer cells and are widely overexpressed in solid tumors [7,129,130,131]. Zhou et al. edited a systematic review of PANDAR deregulation in cancer and proposed it as a biomarker [132]. Yang et al. profiled its expression in 119 AML patients and found PANDAR to be upregulated in various AML subtypes. High levels of PANDAR in patients is significantly associated with higher AML blasts, older patients, and poor karyotypes, but was not correlated with common gene mutations. PANDAR^high^ patients had an adverse prognosis with lower CR rates and shorter OS compared to PANDAR^low^ patients [105]. 

SNHG5 (Small Nucleolar RNA Host Gene 5) is upregulated in the bone marrow and plasma of AML patients compared to healthy controls [106]. SNHG5 upregulation seems to occur more frequently in AML patients with advanced FAB classification and unfavorable cytogenetics. Patients with high plasma SNHG5 expression have significantly shorter OS, and multivariate analysis suggested SNGH5 expression as an independent factor to predict prognosis in AML. 

Dias Beya et al. profiled HOTAIRM1 expression in 241 AML patients with diverse cytogenetic subtypes and observed that HOTAIRM1 is differentially expressed depending on the AML subtypes [109]. The lowest expression level was observed in APL. The most diverse expression of HOTAIRM1 was found in the intermediate risk subgroup (IR-AML) where the prognostic heterogeneity was also the most evident. Amongst this subgroup, the highest levels of HOTAIRM1 were found in NPM1-mutated patients and not significantly associated with other mutations (DNMT3A, IDH1 or IDH2) or any specific clinical feature (age, WBC, FAB subtype). In IR-AML, higher HOTAIRM1 expression is independently associated with shorter OS and DFS, a higher incidence of relapse, mostly for NPM1-mutated patients [109]. These data suggest an oncogenic role of HOTAIRM1 in IR-AML. However, HOTAIRM1 expression levels were neither associated with the probability of attaining CR, nor the frequency of allogeneic hematopoietic stem-cell transplantations (alloHSCT) performed in first CR. They validated these results using arrays from the Leukemia gene Atlas repository [109]. Thus, determination of HOTAIRM1 level provided relevant prognostic information in IR-AML and allowed refinement of risk stratification based on common molecular markers.

CCAT1 and PVT1 (Plasmacytoma Variant Translocation 1) lncRNAs are located on chromosome 8q24, at the proximity of the well-known oncogene C-MYC. CCAT1 and PVT1 upregulation are positively correlated with up regulation of C-MYC in t(8;21) positive AML patients and associated with high-risk criteria, shorter OS and DFS [107]. Moreover; high expression levels of PVT1 and CCAT1 were linked to high minimal residual disease in the t(8;21) positive patients, suggesting their implication in chemoresistance. Interestingly, supernumerary copies of 8q24 chromosomal region is the most common secondary alteration in human APL [133] which might lead to an increase in the number of copies of these lncRNAs. PVT1 is also upregulated in APL patients compared to healthy donors [108]. Both PVT1 and C-MYC expression decreased during ATRA-induced differentiation treatment and knocking either PVT1 or C-MYC reduced the expression of the other.

H19 is upregulated in AML patients compared to healthy donors and correlates with White Blood Cell (WBC) count, intermediate karyotype classifications, FLT3/ITD and DNMT3a recurrent mutations in AML patients [98]. It is an independent prognostic predictor, its overexpression correlating with lower CR rates and shorter OS. The similar results obtained with TCGA and GEO data confirmed the robustness of their study [3,134]. Finally, H19 was proposed as a relapse predictive biomarker, as its expression level is higher at diagnosis, decreased at CR phase but high again on relapse.

#### 4.2.3. LncRNAs Involved in Resistance to Treatments

Significant upregulation of HOTAIR was observed in AML patients at diagnosis compared to healthy donors and its expression is markedly decreased in post-treatment compared to pre-treatment patients [99]. The high expression of HOTAIR is correlated with shorter OS and DFS [62]. Consistent results were observed in additional studies, suggesting its robustness [99,100,101]. High-expression of HOTAIR is also associated with Chronic Myeloid leukemia and bladder cancer with resistance to antileukemic drugs (doxorubicin, immatinib), making this lncRNA a potential therapeutic target to limit drug-resistance [135,136]. 

TUG1 is highly expressed in AML cell lines, and in vitro overexpression promotes cell proliferation and decreases apoptosis rate, suggesting its oncogenic role in AML [102]. Higher expression of TUG1 is also recurrent in AML patients with a monosomal karyotype, FLT3-ITD mutation, poor-risk patients and is correlated with higher WBC count [103]. Luo et al. also demonstrated that AML patients with higher TUG1 expression had shorter OS, lower CR rates and overall response than those with lower TUG1 expression [104]. According to Li et al. TUG1 is also high in doxorubicin-resistant leukemia cells and confers doxorubicin (adriamycin) resistance through PRC2 epigenetic silencing of mir-34a [85]. 

The UCA1 lncRNA also promotes doxorubicin chemoresistance in pediatric AML cases [74]. Indeed, UCA1 expression is upregulated following doxorubicin-based chemotherapy.

Hirano et al. observed that CCDC26, which seems to be restricted to hematopoietic tissue, negatively regulates the expression of KIT. Downregulation of CCDC26 induces a lower growth rate of cells, suggesting that CCDC26 acts as an oncogene to control cell proliferation [110]. Paradoxically, CCDC26-downregulated cells proliferated faster under low serum conditions in vitro and survived longer compared to non KD cells. This observation indicates that suppression of CCDC26 enables leukemia cells to survive and proliferate despite a severe shortage of growth factors. Others suggest resistance of AML cells to anticancer drugs (ATRA treatment) after CCDC26 knock-down by integration of retroviral DNA into the CCDC26 locus, suggesting its putative impact in AML [137]. Paradoxically, Hirano et al. observed that CCDC26 expression is abundant in AML cell lines [110]. Part of CCDC26 is amplified in AML cells harboring double minute chromosomes and the most common copy number alteration found in AML patients appeared in a region within the CCDC26 locus [110,111]. CCDC26 expression level is upregulated at diagnosis and relapse and associated with age, anemia, risk stratification, and remission. Patients with a high CCDC26 expression level had a shorter OS [112].

## 5. From Fundamental to Clinical Research: Incorporation of lncRNAs

### 5.1. Detection and Quantification of lncRNAs In Clinic

Since lncRNAs exhibit high tissue- and disease-specificity, they are ideal candidates for cancer diagnosis and prognosis stratification of patients [138]. In the last years, attention has been dedicated to the detection of such biomarkers in body fluids (Figure 3) [139]. Indeed, lncRNA molecules can be found in different body fluids, such as blood, plasma/serum or urine. They can be derived from apoptotic/necrotic cells, or from living cells which can secrete molecules through exosomes for instance [29,140,141]. Exosomes are released membrane vesicles responsible for trafficking various molecules (proteins, RNAs) throughout biological fluids and are assumed now to be screening biomarker and potential therapeutic target in leukemia [29,141]. 

The techniques currently used to identify and measure lncRNA biomarkers in biopsies are mostly RNA sequencing, microarrays, and qRT-PCR. However, these approaches require many fine-tuning to be used routinely in the clinic. Therefore, there is a need for development of a fast, standardized and clinically applicable tool that would enable the translation of lncRNAs profiling from the bench to the bedside. Bloomfield et al. proposed a probe-assay based on the nCounter platform (providing RNA measurements in a single reaction without amplification) that is designed to produce targeted measurements of prognostic lncRNAs [142]. It is supported by the fact that this technology is already used in an FDA approved assay to quantify the expression of RNA molecules for risk stratification of breast cancer patients [143,144]. To evaluate the robustness of this technique they performed outcome analysis to examine whether the lncRNA score they had previously identified, had retained its prognostic value, and obtained satisfactory correlations [120].

### 5.2. Development of lncRNA-targeted Cancer Therapies

Compared to protein coding transcripts, targeting lncRNAs is challenging. Lack of protein products means that only RNA-based tools are usable. Also, unlike proteins that have specific domains that are easy to target with small molecule drugs, lncRNA conformations are poorly understood, making structure-based strategies difficult to develop. However, as illustrated in Figure 3, several approaches have been proposed to re-establish the homeostatic levels of messenger RNA and could be extended to lncRNAs. 

#### 5.2.1. Targeting Oncogenic lncRNAs

Given their ease of dosage control, low immunogenicity and no danger of genome integration, one of the most explored methods to inhibit up-regulated oncogenic lncRNAs is the delivery of synthetic oligonucleotide-based molecular products (Figure 3) [145]. Prior knowledge of lncRNAs cellular localization is critical for selecting the appropriate strategy to achieve robust lncRNAs modulation [146]. Small interfering RNAs (siRNAs) are double-stranded RNA oligonucleotides, antisense and complementary to target lncRNAs. They induce the degradation of their target by recruiting the RISC (RNA-induced silencing) complex. Very efficient on cytoplasmic targets, these molecules showed a variable success in targeting nuclear lncRNAs. Not yet proposed for the treatment of AML, they have been employed for phase I/II trials for several diseases: PF-04523655, TKM-080301, SYL040012, SYL1001, siG12D-LODER; and others for phase III trials, such as QPI-1002, QPI-1007, and patisiran [147,148,149,150]. SiRNA targeting lncRNA HOTAIR has been shown to suppress the progression of endometrial carcinoma in vivo demonstrating that targeting lncRNA HOTAIR can be a novel therapeutic strategy for endometrial cancer [151]. As HOTAIR is upregulated in AML, the same siRNA strategy could be considered for AML treatment.

Amongst the top upregulated LSC-associated lncRNAs, Bill et al. identified the promising lncRNA DANCR whose knock-down in LSCs resulted in decreased stem-cell renewal and quiescence, suggesting its oncogenic role. The delivery of nanoparticles containing siRNAs against DANCR prolonged the survival of AML mouse model even after serial transplantation [114]. 

Another supportive case of targeting lncRNAs by siRNAs is UCA1, where molecular evidence has established that there is a close relationship between UCA1 and Adriamycin resistance in pediatric AML cases [74]. By using siRNA against UCA1, the authors showed that knock-down of UCA1 plays a positive role in overcoming the chemoresistance of AML cells, through suppressing glycolysis by the miR-125a/HK2 pathway [74]. 

Another synthetic oligonucleotide-based molecular product used to knock-down lncRNA expression has been developed: the Antisense oligonucleotide (ASO). ASO (as GapmeRs) are single-stranded DNA oligonucleotides duplexing by base complementarity their target lncRNAs to promote their degradation by RNase H [152,153]. 

For example, ASO has been used in De Clara et al. study to knock-down the newly identified XLOC_109948 lncRNA. In this study, they showed that low XLOC_109948 expression is correlated with a good prognosis especially for NPM1-mutated AML patients and that its downregulation using ASO in NPM1-mutated AML cells treated with Ara-C or ATRA enhances apoptosis suggesting a role for XLOC_109948 in drug sensitivity [113]. 

Targeting lncRNAs implicated in chemoresistance with interference strategies could improve drug response and outcome. Curna Inc., MiNA Therapeutics, RaNA Therapeutics Inc. and others are taking steps towards the development of oligonucleotide-based strategies [154]. The great challenge is now to improve efficient delivery and long-lasting effects in patients. SiRNAs and ASOs can be modified to overcome their stability and interferon induction effects [155]. To increase intracellular uptakes, various chemical and physical delivery methods are under investigation and were overviewed by Pahle et al. [155] In preclinical studies, these oligonucleotide-based products have been delivered to the heart by intravenous, intraperitoneal, subcutaneous and intracardiac injection or directly administered with catheters. Recombinant viral systems such as adenoviruses, lentiviruses and adeno-associated viruses (AAVs) are commonly used to deliver genetic material such as shRNA and miRNA sponge into a target cell. 

Catalytic degradation is another therapeutic strategy. Ribozymes are single-stranded RNA able to target and cleave RNA in a site-specific manner [156,157]. Anti-VEGFR-1/2 ribozymes are under preclinical studies for the inhibition of liver metastasis obtained in colon cancer models [158]. However, secondary structures of lncRNAs often disrupt the binding efficiency of these molecules to specific targets. 

Aptamers could also provide greater specificity. These single stranded DNA folded into secondary and tertiary structures are able to bind a wide range of molecules, hiding specific-binding structures to limit activity. The chemical structure of aptamers could be modified to enhance their stability and half-life. Aptamer-based -therapeutics are undergoing clinical trials for different indications (non-small cell lung cancer, renal cell carcinoma, and AML) [159]. In 2004, an anti-VEGF aptamer (Eyetech Pharmaceutics/ Pfizer) was approved by the FDA for macular degeneration.

Through their ability to bind proteins to regulate chromatin organization, transcription, and translation, Fatemi et al. proposed innovative therapeutic strategies based on Small molecule Inhibitors able to disrupt lncRNA-protein interactions [160,161]. A study found that 5916 lncRNAs responded to 1262 small molecule drugs [162]. As a matter of fact, LM1070/Branaplam small molecule is under clinical trial for the treatment of Spinal Muscular Atrophy. As it is able to bind the SMN2 pre-mRNA, Branaplam increases SMN2 splicing and translation, antagonizing the disease [163]. This promising study shows that mRNA (and by extension lncRNAs eventhough no studies were reported yet) could be druggable-like proteins. Connelly et al. have identified lncRNAs as promising druggable molecules in the development of new treatments for leukemia [164]. 

#### 5.2.2. Targeting Tumor Suppressor lncRNAs

Re-expression of the specific tumor suppressor lncRNA may be induced by common gene therapy strategies, packaging the whole transcript into viral or non-viral delivery tools. Recombinant viral systems used in gene therapy have been extensively reviewed previously [165]. However, due to the lack of insertion control of viral vectors into the genome, non-viral delivery tools have also been proposed [155,166]. 

Even though its implication as AML clinic target was not proposed yet, HOTAIRM1 lncRNA seems to be a promising candidate in APL treatment. Indeed, as aforementioned, enhanced HOTAIRM1 expression induces degradation of PML-RAR*α* oncoprotein in APL cells and restores the process of myeloid differentiation in those cells [91]. Restoring its expression may be a promising therapeutic target for APL patients.

New Genome editing strategies such as CRISPR/Cas9 are developing very fast and seem to be a powerful tool to knock-in or knock-out lncRNA candidates [167]. Modified CRISPR systems could also generate the substitution of cytidine into uridine in order to correct oncogenic SNPs, knowing that numerous single nucleotide polymorphisms (SNPs) have been associated with potential predictive biomarkers for the risk of cancer, including SNPs in ANRIL, MALAT1, HULC, and PRNCR1 lncRNA [168,169]. 

### 5.3. LncRNAs as Tools

A great challenge in cancer therapy is to selectively kill tumor cells without harming healthy cells. As H19 lncRNA is highly upregulated in several cancers, a plasmid containing diphtheria toxin under the control of H19 regulatory sequences has been designed to selectively target tumor cells which overexpressed H19 compared to normal cells. Diphtheria Toxin is massively produced and kills H19-overexpressing tumor cells selectively. This construct leads to an overall reduction in tumor size and is under clinical trials for ovarian (NCT00826150), pancreatic (NCT00711997) and bladder cancers (NCT00595088) [170]. This construct has been granted FDA Fast Track designation for two Phase III confirmatory studies in September 2015. Since H19 is also upregulated in some AML cases, this strategy could be used to treat AML patients.

Finally, given their tissue specific expression, using the same strategy with the promoter of a myeloid specific lncRNA could lead to the development of a new tool to target selectively myeloid lineage to treat AML patients.

## 6. Conclusions

There is no doubt as to the role of lncRNAs in leukemia development, progression, and drug resistance. However, due to the limited number of studies in AML, their application as therapeutic agents in clinical routine is still at its beginnings. Few results have been replicated across cohorts, probably due to no-standardized sample collection and quantification techniques, but also as a consequence of AML biological complexity, characterized by spatio-temporal relationships between the coding and non-coding genome. Adequate sample sizes, well-designed cohort studies, and validation of the results in independent cohorts are needed to confirm their clinical usefulness. The functional and molecular characterization of lncRNAs is also needed to select the most promising lncRNA targets and to design the best IncRNA therapeutic tool with appropriate efficacy and safety profile. Translating this knowledge into clinical practice still represents a tremendous challenge.

## Figures and Tables

**Figure 1 cancers-11-01638-f001:**
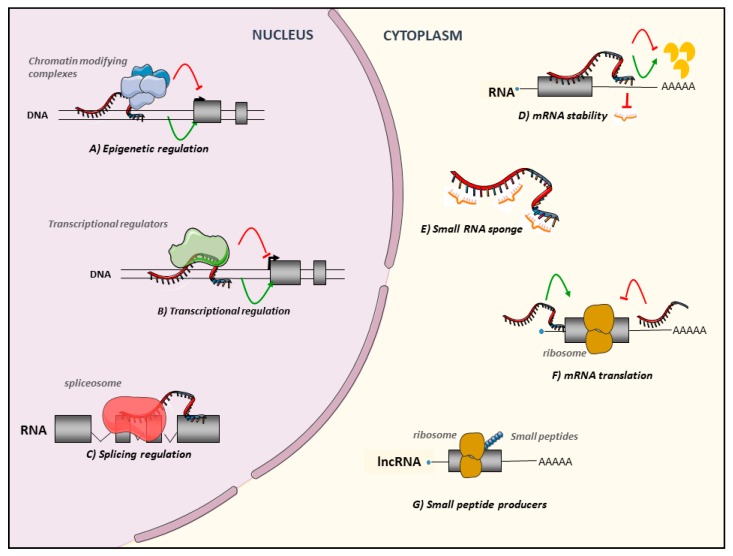
Long Noncoding RNAs: a truly large and multifaced family. Nuclear lncRNAs are implicated in (**A**) Epigenetic regulations, leading to the recruitment of activator/repressor chromatin modifying complexes on their target promoters, (**B**) Transcriptional regulations, guiding or preventing the recruitment of transcription factors on the promoters of their targets or on other active chromatin sites, (**C**) Splicing regulations regulating recruitment of spliceosome partners. Cytoplasmic lncRNAs also affect post-transcriptional steps regulating (**D**) mRNA stability modulating degradation, positively or negatively or acting as (**E**) small regulatory RNA sponges. Finally, they regulate (**F**) mRNA translation. LncRNAs can also be (**G**) small peptide producers.

**Figure 2 cancers-11-01638-f002:**
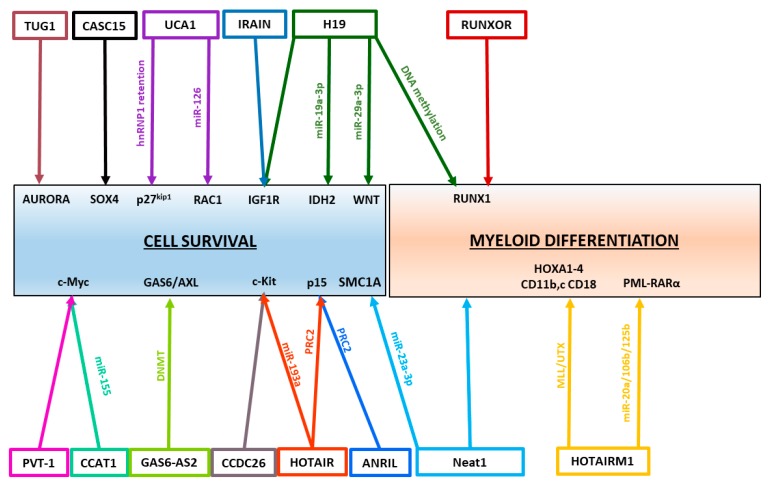
LncRNAs control cell survival and myeloid differentiation leading to AML development. Several examples of lncRNA-controlled pathways are illustrated in this picture, showing known lncRNA targets and how they are regulated.

**Figure 3 cancers-11-01638-f003:**
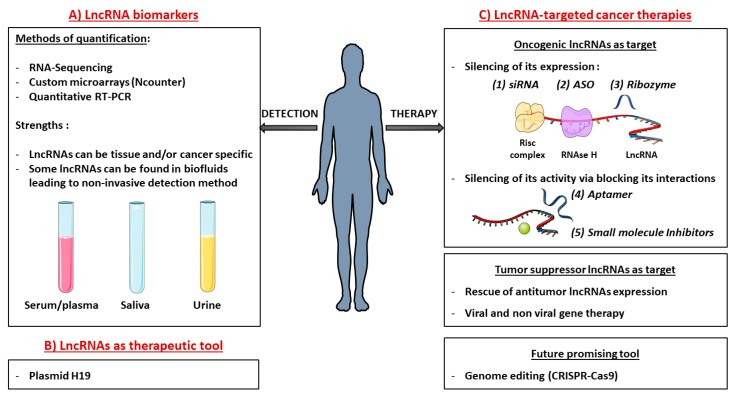
LncRNAs in clinic: (**A**) lncRNA biomarkers. High throughput technologies are used to quantify lncRNAs and highlight their prognostic potential. LncRNAs are promising predictive biomarkers as their expressions can be highly tissue and/or cancer specific. Some of them are even present in biofluids allowing an easy and non-invasive detection method. (**B**) LncRNAs as a therapeutic tool. Highly cancer-specific, lncRNAs can be used as tools to selectively kill tumor cells. For example, a plasmid containing diphtheria toxin under the control of H19 regulatory sequences has been designed to selectively target tumor cells which overexpressed H19 compared to normal cells. Diphtheria toxin is massively produced and selectively kills H19-overexpressing tumor cells. (**C**) LncRNAs targeted cancer therapies. Oncogenic LncRNAs as target: silencing of its expression: (**1**) Small interfering RNA (siRNA) are double-stranded RNA oligonucleotides antisense and complementary to target lncRNA sequences. They induce degradation of their target by recruiting the RISC (RNA-induced silencing) complex. (**2**) An antisense oligonucleotide (ASO) is a single-stranded DNA oligonucleotide that is complementary to the target RNA and able to induce its degradation by recruiting RNase H. (**3**) Ribozymes (Ribonucleic acid Enzymes) are single-stranded RNA. By adopting specific conformations, they are able to bind RNA targets and catalyze their degradation. Oncogenic LncRNAs as target: silencing their activity: Small molecules are also able to hide partners’ interaction sites (DNA, RNA, proteins) to suppress their activities. (**4**) Aptamers are single stranded DNA folded into secondary and tertiary structures that can bind specific structural regions of the target lncRNAs. (**5**) Small molecule Inhibitors can also disrupt lncRNA interactions. Tumor suppressor lncRNAs as target: Rescue of its expression: It can be provided by common gene therapy strategies, packaging the whole transcript into viral or non-viral delivery tools. Future promising tool: (**J**) New Genome editing strategies such as CRISPR/Cas9 are developing very fast to knock-in/out lncRNA candidates.

**Table 1 cancers-11-01638-t001:** Oncogenic lncRNAs in Acute Myeloid Leukemia.

LncRNA	Clinical Significance	Mechanisms of Actions	Functions in Leukemogenesis	Ref
H19	—Upregulated in AML —Correlated with WBC count/karyotypic classifications/*FLT3-ITD* and *DNMT3a* mutations/chemotherapy response/OS; high at diagnosis/relapse	—Maternal imprinting of *IGF2* gene —miR-19a-3p and 29a-3p sponges —Promoter methylation of hematopoietic transcription factors (*RUNX1/SP1*)	—Sustain Adult HSC quiescence, leukemic cell proliferation —Limit apoptosis	[96,97] [98]
HOTAIR	—Upregulated in AML —Correlated with OS/DFS/drug resistance —Downregulated after treatment	—Repress *p15* expression (PRC2) —Regulate *c-KIT* expression by sponging miR-193a	—Sustain Cell growth —Inhibit apoptosis	[62,63,64] [99,100,101]
TUG1	—Upregulated in AML with monosomal karyotype/FLT3-ITD mutation/ poor-risk patients —Correlated with high WBC count/shorter OS/lower rate of CR —Implicated in Adriamycin resistance	Target *Aurora Kinase*	—Promote cell proliferation —Inhibit Apoptosis —Doxorubicin resistance	[85] [102,103,104]
UCA1	—Upregulated in CN-AML with dominant negative C/EBPα —Upregulated in Adriamycin resistant pediatric AML cases	—Inhibit p27kip1 translation by titrating hnRNPI factor —miR-125a and miR-126 sponges	—Sustain AML cell proliferation, migration, invasion —Inhibit Apoptosis —Doxorubicin Resistance	[73,74,75,76]
PANDAR	—Upregulated in AML —Associated with higher AML blasts/older patients/poor karyotypes/lower OS and CR			[105]
RUNXOR	—Upregulated in t(8;21) AML and after ARA-C treatment	—Intrachromosomal loop between *RUNX1* promoter and enhancers —Interacts with PRC2 and RUNX1 protein to regulate *RUNX1* expression		[70]
SNHG5	—Upregulated in AML patients —Associated with advanced FAB classification and unfavorable cytogenetics, Shorter OS	—miR-205-5P sponge		[106]
ANRIL	—Upregulated in AML at diagnosis and downregulated after CR	—Silencing of *p15INK4B* by scaffolding PRC2 —Regulate expression of *AdiR1*	—AML cell maintenance —Implicated in Glucose metabolism	[57]
PVT-1	—Upregulated in APL and t(8;21) AML —Associated with high-risk criteria/shorter OS and DFS	—miR-200 sponge: *c-MYC* regulation	—Sustain proliferation of promyelocytes	[107] [108]
CCAT1	—Upregulated in AML patients (mostly in M4-M5 subtypes)	—miR-155 sponge: *c-MYC* regulation	—Repress monocytic differentiation —Promote cell growth	[107]
HOTAIRM1	—High expression associated with shorter OS and DFS/higher incidence of relapse in IR-AML, mostly in *NPM1* mutated patients	—Activate expression of proximal *HOXA1-4, HOXA4, CD11b, c,* and *CD18* genes —miR-20a/106b and miR-125b sponges	—Regulate myeloid differentiation, cell cycle, and autophagy pathways	[65,66,67] [109]
CCDC26	—Upregulated at diagnosis and relapse —Associated with age, anemia, risk stratification, remission, shorter OS	Repress *c-Kit* expression	—Sustain AML cell proliferation —Resistance to anticancer drugs (ATRA treatment)	[110,111,112]
XLOC_ 109948	—Low expression indicates a good prognosis, especially for NPM1-mutated AML patients		Apoptosis/ drug resistance	[113]
DANCR			—Knock-down in LSCs resulted in decreased stem-cell renewal and quiescence	[114]

**Table 2 cancers-11-01638-t002:** Tumor Suppressor lncRNAs in Acute Myeloid Leukemia.

LncRNA	Clinical Significance	Mechanisms of Actions	Functions in Hematopoiesis	Ref
IRAIN	Downregulated in AML cell lines; Lower in patients with high-risk AML —Associated with higher WBC counts/shorter OS and DFS/Refractory response to chemotherapies/relapse	—Intrachromosomal enhancer/promoter loop of *IGF1R* gene	—Inhibit tumor cell migration	[71,72]
NEAT1	—Downregulated in AML patients with *PML-RARα* translocation	—miR-23a-3p sponge	—Regulate myeloid differentiation —Inhibit AML cell proliferation, induce cell-cycle arrest and apoptosis	[93,94,95]
MEG3	—Hypermethylation of *MEG3* promoter in AML —Downregulated in AML —Associated with longer OS and DFS	—Positive regulation of *p53* expression	—Regulate cell cycle, apoptosis	[115,116,117,118]
CASC15	—Upregulated in RUNX1-rearranged AML —Highest level found in AML with the t(8;21) translocation —Associated with a good prognosis	—*CASC15* activates expression of *SOX4* gene, by regulating the activity of the YY1 transcription factor	—Increase apoptosis and myeloid cells number	[119]

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
