# Peer review of "Long Noncoding RNAs in Acute Myeloid Leukemia: Functional Characterization and Clinical Relevance"

_cancers, 2019, doi:10.3390/cancers11111638_

Round 1
Reviewer 1 Report
This is a comprehensive review of the role of long noncoding RNAs in AML and will be a helpful resource for the field. The tables are particularly helpful. In addition, the writing is clear, with only a few minor corrections suggested, as listed below.
line 91- it should read "non-invasive", not "no-invasive" line 161- the sentence "The following aim is to validate..." seems out of place. line 186- should read "Wnt", not "wint"Author Response
This is a comprehensive review of the role of long noncoding RNAs in AML and will be a helpful resource for the field. The tables are particularly helpful. In addition, the writing is clear, with only a few minor corrections suggested, as listed below.
We really would like to thank the reviewer for his nice comment on this manuscript. We addressed all of the minor corrections that he suggested.
line 91- it should read "non-invasive", not "no-invasive"
We corrected it, see line 117
line 161- the sentence "The following aim is to validate..." seems out of place.
Indeed, we remove it from the text.
line 186- should read "Wnt", not "wint"
We corrected it, see line 227
Reviewer 2 Report
The authors has written the article elegantly and covered all the areas about lncRNAs and its role in AML. Specifically focusing on lncRNA that are linked to leukemic cells with a focus on their prognostic and therapeutic potential uses.
Comment:
There are several mechanisms described in Fig.1 for the transcriptional regulation by lncRNAs. But if authors can atleast mention or revise using the following article to show that lncRNAs NEAT1 binds to promoter region and MALAT1 binds to 3'UTR in regulating the transcription.
'The long noncoding RNAs NEAT1 and MALAT1 bind active chromatin sites' in Mol Cell. 2014 Sep 4;55(5):791-802.
I hope this will help in readers an indepth understanding of the lncRNA regulation during transcription.
Author Response
The authors has written the article elegantly and covered all the areas about lncRNAs and its role in AML. Specifically focusing on lncRNA that are linked to leukemic cells with a focus on their prognostic and therapeutic potential uses.
Comment:
There are several mechanisms described in Fig.1 for the transcriptional regulation by lncRNAs. But if authors can at least mention or revise using the following article to show that lncRNAs NEAT1 binds to promoter region and MALAT1 binds to 3'UTR in regulating the transcription.
'The long noncoding RNAs NEAT1 and MALAT1 bind active chromatin sites' in Mol Cell. 2014 Sep 4;55(5):791-802.
I hope this will help in readers an indepth understanding of the lncRNA regulation during transcription.
We really would like to thank the reviewer for his nice comments on this manuscript.
As suggested, we now mentioned this interesting article by adding a sentence and the reference lines 93-94
Reviewer 3 Report
The presented manuscript" Long noncoding RNAs in Acute Myeloid Leukemia: Functional characterization and Clinical relevance" is an interesting review of lncRNAs associated with cancer, and their application as therapeutic agents. The authors provide a long catalog of examples, with numerous references which is the strength of the paper. However, this length creates a weakness in the end of the paper, since authors do not really replace many given examples in the context of AML studies. This weakness leads the readers to see this paper more as a paper of lncRNAs known in cancer, than a paper focused on AML. Finally, references contain many missing information and are not well-formatted. These issues could be easily fix by the authors.
General Comments:
Part 1 and 2: Good introduction about AML and lncRNAs. Some typo issues (see: minor changes)
I am wondering if you could make a figure showing the general gene/protein pathway associated with AML, and where are located the known lncRNAs associated with AML described in part 3, within this pathway.
Part 3:Long catalog of lncRNAs associated with AML. Maybe you could write a shorter list and add one or two paragraphs discussing common pathway/function/actions of these examples (and keeping only full list for tables 1 and 2).
Line 289: Can you move this part with the previous part about HOTAIR
Following that, can you discuss/give your point of view about the difference/common actions of HOTAIR/HOTAIRM1 and about the HOX genes family in cancer.
Part 4: Weakest part: you give four examples of lncRNAs, giving information about their associated diseases, but finishing most of your examples by "it will be interesting to investigate it in AML patients". You don't really explain why? You just give us four examples not yet described in AML. I would like to know why these particular examples were chosen. Which effect do you expect to observe in AML patients related to the expression or absence of these lncRNAs? As I said, you have to put these examples in the context of AML studies.
Part 5. It is nice to give examples of lncRNA therapies developed, but again, can you add more strength in this part putting these examples in the context of AML. Your paper is about AML. Without any discussion about potential examples/hypothesis of therapeutic targets in AML patients, you could write this part easily for any kind of cancer. This part is just before your conclusion in which you say "there are limited numbers of studies in AML". So, you should take examples of lncRNAs associated with AML that you described in part 3 and make comparison with known therapeutic studies to propose some ideas for AML studies. Following this recommendation, the quality of your paper will be really improved, beyond a simple catalog of known studies.
Minor changes:
Usually, plural for lncRNA are ended by "s": lncRNAs but I let the editors to choose. But reading will be easier with a "s". Also, many gene names are not italicized. fix them.
Line 80: Please define HSC for non-cancer readers.
Line 95- "It is a Prostate-specific" (remove uppercase)
Line 99: move references after the dot
Line 102: remove extraspace before reference
Line 119: Affymetrix is a company, not a molecular tool, so please specify which technologies they used to study lncRNA expressions.
Line 120-121: t(8 ;16) or t(8;16) ?
Line 112: one comma missing between IDH2 ASLX1
Line 143: remove extraspace before reference
Line 181: please, write "the tumor suppressor retinoblastoma (RB)"
Line 183: References are not merge in the same [ ]
Line 199: fix "TCGA and GEO databases". I recommend adding references for both databases.
Line 263: Maybe the reference will be better at the end of sentence L260?
Line 301: remove the uppercase "Intermediate risk subgroup" and keep after (IR-AML)
Line 250: remove extraspace before reference
Lines 277-279 and table 1: Your reference 90 does not really speak about PVT1. They just notified that Pvt1 and Myc are in this region, with perhaps Myc involved. So you should remove this reference in your table 1 and you should move this reference like this in your sentence:
"Supernumerary copies of this chromosomal region is the most common secondary alteration in human APL[90] which might lead to an increase in the number of copies of PVT1." Perhaps "which might lead to an increase..." could be a better English to reflect the fact that this is your hypothesis and not the paper result. Like this, you provide more strength to your text with your own arguments.
L320: missing space after reference [98]
Table 1 and 2: several Uppercase missing - What are Up? Down? what is Higher HOTAIRM1? sometimes you write upregulated, downregulated. please use same terminology for all.
Line 379. you don't need uppercases for "lesson from normal hematopoiesis
Line 380: same comment.
Line 434: "their expressions"
Line 436-439: reference missing
Line 438, 525, 527: "Diptheria toxin"
Line 493: remove extraspace before reference
Figure 2. A) LncRNA biomarkers C) Viral and non-viral gene therapy
References
Please check the journal recommendations about number of authors, etc...
Some references are not completed or not well formatted: Please be careful, some have missing information, sometimes date in bold, sometimes not...
Author Response
The presented manuscript" Long noncoding RNAs in Acute Myeloid Leukemia: Functional characterization and Clinical relevance" is an interesting review of lncRNAs associated with cancer, and their application as therapeutic agents. The authors provide a long catalog of examples, with numerous references which is the strength of the paper. However, this length creates a weakness in the end of the paper, since authors do not really replace many given examples in the context of AML studies. This weakness leads the readers to see this paper more as a paper of lncRNAs known in cancer, than a paper focused on AML. Finally, references contain many missing information and are not well-formatted. These issues could be easily fix by the authors.
We would like to thank the reviewer for all of their fruitful comments that will greatly improve the quality of the manuscript.
General Comments:
Part 1 and 2: Good introduction about AML and lncRNAs. Some typo issues (see: minor changes)
I am wondering if you could make a figure showing the general gene/protein pathway associated with AML, and where are located the known lncRNAs associated with AML described in part 3, within this pathway.
According to your advice, and because cell survival and myeloid differentiation are the major deregulated mechanisms in AML, we added a figure 2 representing how each lncRNA is implicated in these deregulations. We segregated lncRNA depending on their implication in cell survival or myeloid differentiation. And we tried to incorporate associated pathways/mechanisms of action when it is described in the literature. We hope to have met your expectations with this figure 2.
Part 3:Long catalog of lncRNAs associated with AML. Maybe you could write a shorter list and add one or two paragraphs discussing common pathway/function/actions of these examples (and keeping only full list for tables 1 and 2).
According to your advice, we tried to avoid this long catalog of lncRNAs and we choose some example to illustrate the main mechanisms of action (See Part 3 “Regulatory Roles of lncRNAs”). Some of them are now just described in the part 4 to illustrate the prognostic potential of lncRNAs (See Part 4.2 Prognostic lncRNA biomarker candidates).
Line 289: Can you move this part with the previous part about HOTAIR. Following that, can you discuss/give your point of view about the difference/common actions of HOTAIR/HOTAIRM1 and about the HOX genes family in cancer.
According to your advice, we linked paragraphs on HOTAIR and HOTAIRM1 and described them as modulators of HOX gene expression with opposite mechanisms (Line 154- 176)
Part 4: Weakest part: you give four examples of lncRNAs, giving information about their associated diseases, but finishing most of your examples by "it will be interesting to investigate it in AML patients". You don't really explain why? You just give us four examples not yet described in AML. I would like to know why these particular examples were chosen. Which effect do you expect to observe in AML patients related to the expression or absence of these lncRNAs? As I said, you have to put these examples in the context of AML studies.
We decided to delete this part of the article, because of the lack of information about these lncRNA in AML. Indeed, their deregulations lead to evident hematopoietic defects, but not necessarily AML.
Part 5. It is nice to give examples of lncRNA therapies developed, but again, can you add more strength in this part putting these examples in the context of AML. Your paper is about AML. Without any discussion about potential examples/hypothesis of therapeutic targets in AML patients, you could write this part easily for any kind of cancer. This part is just before your conclusion in which you say "there are limited numbers of studies in AML". So, you should take examples of lncRNAs associated with AML that you described in part 3 and make comparison with known therapeutic studies to propose some ideas for AML studies. Following this recommendation, the quality of your paper will be really improved, beyond a simple catalog of known studies.
We would like to thank the reviewer for this comment. We rewrote this part (Part 5) to include some example in AML or some ideas of therapy in the context of AML.
Minor changes:
All of the minor changes have been addressed.
Usually, plural for lncRNA are ended by "s": lncRNAs but I let the editors to choose. But reading will be easier with a "s". Also, many gene names are not italicized. fix them.
Line 80: Please define HSC for non-cancer readers.
Line 95- "It is a Prostate-specific" (remove uppercase)
Line 99: move references after the dot
Line 102: remove extraspace before reference
Line 119: Affymetrix is a company, not a molecular tool, so please specify which technologies they used to study lncRNA expressions.
We replace Affymetrix by microarray. See line 265
Line 120-121: t(8 ;16) or t(8;16) ?
Line 112: one comma missing between IDH2 ASLX1
Line 143: remove extraspace before reference
Line 181: please, write "the tumor suppressor retinoblastoma (RB)"
After the reorganization of the paragraph, we decided to delete this information from the text.
Line 183: References are not merge in the same [ ]
Line 199: fix "TCGA and GEO databases". I recommend adding references for both databases.
We added references for both. Line 378
Line 263: Maybe the reference will be better at the end of sentence L260?
Line 301: remove the uppercase "Intermediate risk subgroup" and keep after (IR-AML)
Line 250: remove extraspace before reference
Lines 277-279 and table 1: Your reference 90 does not really speak about PVT1. They just notified that Pvt1 and Myc are in this region, with perhaps Myc involved. So you should remove this reference in your table 1 and you should move this reference like this in your sentence:
"Supernumerary copies of this chromosomal region is the most common secondary alteration in human APL[90] which might lead to an increase in the number of copies of PVT1." Perhaps "which might lead to an increase..." could be a better English to reflect the fact that this is your hypothesis and not the paper result. Like this, you provide more strength to your text with your own arguments.
According to your advice, we have rearranged this paragraph. (Line 363-373) We have changed the wording; we have also removed the reference in the table 1.
L320: missing space after reference [98]
Table 1 and 2: several Uppercase missing - What are Up? Down? what is Higher HOTAIRM1? sometimes you write upregulated, downregulated. please use same terminology for all.
We changed “up” by “upregulated”, “down” by “downregulated”. We wrote upregulated or downregulated only when it was clearly written in the publications that the lncRNA expression is deregulated compared to normal. We keep “high or low” expression, when they talk about adverse expression among AML subtypes.
Line 379. you don't need uppercases for "lesson from normal hematopoiesis
Line 380: same comment.
Line 434: "their expressions"
Line 436-439: reference missing
Line 438, 525, 527: "Diptheria toxin"
Line 493: remove extraspace before reference
Figure 2. A) LncRNA biomarkers C) Viral and non-viral gene therapy
References: Please check the journal recommendations about number of authors, etc... Some references are not completed or not well formatted: Please be careful, some have missing information, sometimes date in bold, sometimes not...
We are very sorry about this mistake. We fixed it.
Reviewer 4 Report
Dear Authors,
I appreciate your review on "role of Long noncoding RNAs in Acute Myeloid Leukemia"
However; it would be worth reading if you add the following to the manuscript
Characteristics, Biogenesis and Classification oflncRNA, Transcriptional and Epigenetic Regulation of lncRNA, lncRNA versus miRNA and the roles of exosomes add more details for diagnosis applications and future directionsThanks,
LR
Author Response
Dear Authors,
I appreciate your review on "role of Long noncoding RNAs in Acute Myeloid Leukemia"
However; it would be worth reading if you add the following to the manuscript
Characteristics, Biogenesis and Classification of lncRNA, Transcriptional and Epigenetic Regulation of lncRNA, lncRNA versus miRNA and the roles of exosomes
add more details for diagnosis applications and future directions
We would like to thank the reviewer for all of their fruitful comments that will greatly improve the quality of the manuscript.
We have added several paragraphs providing more details on the characteristics, biogenesis and classification of lncRNAs. We also mentioned some interesting revues that detail these different aspects. (See Part 2)
We also included more details about lncRNAs as microRNA sponges (line 98-101)
We also added informations about exosomes and mentioned references about exosomes and their usefulness in leukemia. (lines 72-73 and lines 422-424)
Round 2
Reviewer 3 Report
This version of the paper " Long noncoding RNAs in Acute Myeloid Leukemia : Functional characterization and Clinical relevance" is much better, well organized and more complete. Authors followed reviewers' suggestions and now the paper is ready to be published, and will be very interesting for the lncRNA/cancers community.
Just 3 typos:
L67: one space missing after reference [25].
L154: one dot is missing at the end of the sentence
L435: Diphteria toxin (no uppercase for toxin)